# *Candida albicans* Interactions with Mucosal Surfaces during Health and Disease

**DOI:** 10.3390/pathogens8020053

**Published:** 2019-04-22

**Authors:** Spyridoula-Angeliki Nikou, Nessim Kichik, Rhys Brown, Nicole O. Ponde, Jemima Ho, Julian R. Naglik, Jonathan P. Richardson

**Affiliations:** 1Protein Phosphorylation Laboratory, The Francis Crick Institute, London NW1 1AT, UK; spyridoula.nikou@crick.ac.uk; 2Centre for Host-Microbiome Interactions, Faculty of Dentistry, Oral & Craniofacial Sciences, King’s College London, London SE1 1UL, UK; nessim.kichik@kcl.ac.uk (N.K.); rhys.brown@kcl.ac.uk (R.B.); nicole.ponde@kcl.ac.uk (N.O.P.); jemima.ho@kcl.ac.uk (J.H.); julian.naglik@kcl.ac.uk (J.R.N.)

**Keywords:** *Candida albicans*, commensal, pathogen, fungus, mucosal infection, microbiota

## Abstract

Flexible adaptation to the host environment is a critical trait that underpins the success of numerous microbes. The polymorphic fungus *Candida albicans* has evolved to persist in the numerous challenging niches of the human body. The interaction of *C. albicans* with a mucosal surface is an essential prerequisite for fungal colonisation and epitomises the complex interface between microbe and host. *C. albicans* exhibits numerous adaptations to a healthy host that permit commensal colonisation of mucosal surfaces without provoking an overt immune response that may lead to clearance. Conversely, fungal adaptation to impaired immune fitness at mucosal surfaces enables pathogenic infiltration into underlying tissues, often with devastating consequences. This review will summarise our current understanding of the complex interactions that occur between *C. albicans* and the mucosal surfaces of the human body.

## 1. Introduction

The human body provides a multitude of disparate and challenging niches to colonising microbes, including the mucosal surfaces of the oropharyngeal, gastrointestinal and vaginal tracts. *Candida albicans* has evolved to persist at mucosal surfaces [1,2] as a benign component of the microbiota, and is superbly adapted to life in the host as a commensal organism, particularly in the gastrointestinal tract. While frequently colonised by *C. albicans*, mucosal surfaces nevertheless play a vital role in host protection and are crucial for the appropriate initiation and coordination of innate immune responses during infection. However, under circumstances where host immunity is impaired, *C. albicans* can transition from a harmless commensal to a pathogen capable of breaching mucosal barriers, causing deep seated invasive and life-threatening disseminated infection.

Coevolution of *C. albicans* with the human host has resulted in both organisms acquiring the means to adapt to one another. This coevolutionary “coin” comprises continual fungal adaptation to the host on one side, and a perpetual evolution of the host immune response to the fungus on the other. While the majority of scientific studies have focused on *C. albicans* hyphae and associated virulence factors, both yeast and hyphal morphologies contribute to fungal persistence in the host. A physical interaction between *C. albicans* and a mucosal surface is a necessary requirement that precedes commensal colonisation and pathogenic infiltration.

This review article will examine the numerous events that transpire during the interaction of *C. albicans* with the mucosal surfaces of the human body; mechanistic and structural aspects of adhesion will be considered together with the processes of epithelial internalisation, the role of secreted host and fungal factors, and the acquisition of essential micronutrients.

## 2. Adhesion of *C. albicans* to the Epithelium

Adhesion of *C. albicans* to a mucosal surface is an essential requirement for persistence in the host, whether it be as a commensal or a pathogen [3,4]. The mucosal surfaces of the body are covered with a protective coating of mucous which must be traversed in order for *C. albicans* to attach itself to underlying epithelial cells. Indeed, adhesion of *C. albicans* to buccal epithelial cells is reduced in the presence of purified mucin [5]. The majority of initial contact between *C. albicans* and the host is thought to involve yeast, with germ tube and hypha formation occurring after initial contact. Yeast cells have evolved a number of strategies to ensure successful adherence to the host epithelium. Initial interactions between *C. albicans* and epithelial cells rely on a number of attractive and repulsive forces, including van der Waals forces and hydrophobic interactions [6]. While these passive forces are by no means the predominant mechanism required for long-term fungal adhesion, they are nevertheless vital for the initiation of adherence. *C. albicans* and epithelial cells are considered to possess a net negative charge, implying a degree of electrostatic repulsion that opposes physical association [7]. Successful contact between *C. albicans* and epithelial cells is thus dependent on the sum of attractive forces outweighing those which promote cellular repulsion [8]. Adhesion of yeast cells to epithelial cells positively correlates with the expression of cell surface hydrophobins [9,10], while cetylpyridinium chloride-induced reduction of cell surface hydrophobicity correlates with decreased adhesion [11].

Once attached to the mucosal surface, the adhesion of *C. albicans* is further strengthened by numerous interactions with components of the host extracellular matrix. *C. albicans* yeast can bind to human fibronectin [12,13], proline-rich salivary proteins [14] and carbohydrates that facilitate adhesion to human oesophageal epithelial cells [15] and buccal epithelial cells in vitro [16].

However, the greatest contribution to fungal adhesion is conferred by the adhesins. The best studied of the *C. albicans* adhesins are the agglutinin-like sequence (Als) family of proteins; consisting of eight members (Als1p–Als7p and Als9p) that are glycosylphosphatidylinositol (GPI)-linked to the β-1,6-glucans of the fungal cell wall (reviewed in [17]). Als5p mediates the initial adhesion of *C. albicans* yeast cells to human buccal epithelial cells, and to patches of threonine, serine, and alanine residues within fibronectin, type IV collagen and laminin [18,19,20]. A conserved tandem repeat region within Als5p facilitates adhesion to numerous epithelial ligands and promotes yeast-to-yeast cell aggregation [21,22]. The conserved amyloid forming sequences of Als5p are implicated in a transition away from overt pathogenicity towards gastrointestinal commensalism in vivo [23] and are important for coordinating the clustering of adhesins on the *C. albicans* fungal cell wall, facilitating continued fungal adhesion to the host by increasing the likelihood of epithelial cell ligands rebinding to nearby adhesins should detachment occur [24,25]. 

Expression of *ALS* genes differs according to fungal morphology and body site. *ALS1–5* and *ALS9* were consistently upregulated in a reconstituted human buccal epithelium model of mucocutaneous candidiasis, while *ALS6* and *ALS7* exhibited variable expression. In contrast, *ALS1*, *ALS2*, *ALS3* and *ALS9* were expressed frequently in clinical specimens of vaginal fluid, while transcripts from *ALS4* and *ALS5* were detected less frequently [26]. This differential expression not only suggests a degree of functional redundancy between Als adhesins but suggests specific roles for particular adhesins at different mucosal sites [27]. The factors that contribute to the adhesion of *C. albicans* yeast to epithelial cells are depicted in Figure 1A.

Of the Als proteins, Als3p is key in the adherence of *C. albicans* hyphae to epithelial cells and the subsequent invasion of host cells. *ALS3* is upregulated during infection of oral and vaginal epithelial cells [26,28,29,30] while blocking Als3p or preventing *ALS3* expression causes a significant reduction in epithelial adhesion [17,31]. 

However, the role of certain Als proteins in *C. albicans*–epithelial interaction is less clear. For instance, while deletion of *ALS1* caused a significant decrease in fungal adhesion to murine tongue tissue ex vivo [32] reports also suggest that Als1p plays only a minor role in adhesion, particularly when compared with Als3p [33]. Similarly, while deletion of *ALS2* decreased the adhesion of *C. albicans* to a reconstituted model of human oral epithelium, deletion of *ALS4* in the same model had no effect [34]. 

Given the degree of functional redundancy observed between adhesins and the inherent complexity associated with adhesion of *C. albicans* to a diverse range of host tissues, it is perhaps unsurprising that the process of adhesion is subject to numerous influential forces. Indeed, several antiadhesive factors have been identified which may serve to fine-tune the process of adhesion. Yeast wall protein 1 (Ywp1p/Pga24p) is a yeast-specific GPI-linked glycoprotein that is highly expressed during yeast but not hyphal growth [35]. Deletion studies demonstrate that yeast cells lacking *YWP1* are more adhesive [36,37], suggesting a role for Ywp1p in the dispersal of yeast, which may facilitate dissemination [37]. Deletion of *ALS5*, *ALS6* and *ALS7* increase adhesion to epithelial cells, suggesting that these proteins also possess antiadhesive properties [38]. The major factors that contribute to the adhesion of *C. albicans* hyphae to epithelial cells are depicted in Figure 1B.

## 3. Structural and Mechanistic Aspects of Adhesion to the Epithelial Surface

Adhesion is a complex process that results from the simultaneous interaction of fungal cell wall components with the biomolecules present on the surface of the host cell membrane. While the fungal cell wall is composed of proteins and carbohydrates (chitin, glucan and mannans), the composition of proteins and sugars differs between yeast and hyphal morphologies. In hyphae, adhesins and proteins involved in cell wall synthesis are upregulated resulting in stronger adhesion when compared to yeast [39].

While most studies investigating *C. albicans* adhesion have used genetic approaches, relatively little is known about the structural basis of adhesion to epithelial cells. Hyphal regulated 1 (Hyr1p), hyphal wall protein 1 (Hwp1p) and the Als proteins share a similar arrangement of domains; a folded N-terminal domain responsible for protein–protein/ligand interaction with the host followed by a serine/threonine-rich variable tandem domain of low complexity and a C-terminal peptide sequence that covalently binds to a GPI lipid anchor in the fungal cell wall [40]. However, no structural information is available for Hyr1p and Hwp1p adhesins at present. 

The N-terminal domain of Hwp1p exhibits 40–50% amino acid sequence identity with the central and N-terminal domains of the human small proline-rich (SPR) family of proteins which are substrates of the transglutaminase (TGase) family of enzymes. At the interface between the *C. albicans* cell wall and the epithelial cell membrane, host TGases cross-link Hwp1p to lysine residues within the proteins present on the epithelial surface [41]. 

Our understanding of the structure–function relationship that enables adhesion to the host is most advanced for the Als family of adhesins [42,43], which recognise and bind to a variety of structurally unrelated ligands. Such flexibility enables *C. albicans* to attach to surfaces with different compositions including bacterial and mammalian cells and to abiotic surfaces such as medical devices [44]. Numerous Als binding proteins have been identified including Streptococcus gordonii SspB, bovine serum albumin, gp96, collagen, laminin, casein, fibrinogen, human and equine ferritin and the human epidermal growth factor receptors EGFR/HER1 and HER2, fibronectin, E-cadherin and N-cadherin [20,28,29,45,46,47,48]. 

An X-ray crystal structure of the N-terminal domain of an allelic variant of Als9p (designated Als9-2) complexed with the C-terminal region of human fibrinogen-γ reveals that the Als adhesins recognise flexible C-terminal peptides of up to six amino acids which are accommodated in an extended conformation inside a peptide binding cavity (PBC) [49]. A model describing how Als adhesins recognise and bind structurally unrelated ligands has been proposed, in which the wide, flat PBC located in the N-terminal domain of the Als adhesin can accommodate a wide range of 6-mer peptide ligands [50]. Once located in the PBC, the peptide backbone of the ligand forms an extensive network of hydrogen bonds with the residues of the PBC and a salt bridge is formed between the C-terminal carboxylic acid of the incoming peptide and the positively charged side chain of a conserved lysine located at the end of the PBC. However, most transmembrane-spanning proteins typically have their free C-termini located on the intracellular face of the plasma membrane, rendering them inaccessible to the PBC of Als adhesins. To mitigate this issue, a mechanism was proposed in which secretion of fungal proteinases enables limited proteolytic digestion of the extracellular regions of membrane-associated host proteins, liberating free C-termini on the extracellular face of the plasma membrane that may subsequently be recognised by the PBC [51]. However, it must also be noted that Als1p and Als5p are capable of binding to peptides displaying free N- or C-termini [20,52].

The Als family of proteins also contain an amyloid forming region (AFR), located in the N-terminal domain. Als proteins can interact with each other via their AFR motifs to form large molecular weight clusters of Als proteins, termed nanodomains [53,54], which are implicated in the aggregation of *C. albicans* hyphae, biofilm formation and cell adhesion. Furthermore, the interaction between soluble regions of AFR motifs in Als5p results in the formation of amyloid fibers [53]. Analysis of AFR motifs by nuclear magnetic resonance (NMR) spectroscopy demonstrate that Als proteins interact with each other via their AFR domains to form Als aggregates in the absence of ligands. However, in the presence of ligands that bind to the PBC these interactions are disrupted. These findings highlight that ligand binding and Als–Als aggregation are mechanistically distinct. Disruption of the PBC but not the AFR results in decreased fungal adhesion to the commensal bacterium *S. gordonii*, suggesting that the PBC of Als3p plays a primary role in mediating the interaction between *C. albicans* and microbial cells whereas the AFR motif is necessary for Als–Als clustering in the absence of Als ligands [50,51].

## 4. The Commensal Relationship with the Host

*C. albicans* is exquisitely adapted to life in the host as a commensal, particularly in the gastrointestinal tract. This continually evolving commensal relationship between *C. albicans* and the human body is typified by asymptomatic carriage. While commensal colonisation of mucosal surfaces is often associated with yeast rather than hyphae, both morphologies have been observed to colonise the murine gastrointestinal tract [55].

Mouse models of gastrointestinal colonisation demonstrate that *C. albicans* can stably colonise the gastrointestinal tract in a little as three days [56]. The mitogen-activated protein kinase Hog1p confers adaptation to oxidative and osmotic stress, and is essential for gastrointestinal colonisation [57] while numerous transcriptional regulators, including Tye7p, Lys144p, Hms1p, Rtg1p, Rgt3p and ORF19.3625, play a significant role in colonisation [58]. 

Recently, significant advances in our understanding of the fine balance between beneficial and detrimental antifungal immune responses have been made. Commensal colonisation of the host intestine drives the expansion of systemic Th17 CD4^+^ T cells that, together with IL-17-responsive neutrophils, protect against invasive infection [59]. However, colonisation was observed to exacerbate susceptibility to allergic airway inflammation [59]. Indeed, the process of intestinal inflammation drives the expansion of *C. albicans*-specific Th17 cells and a pool of Th17 cells that exhibit cross-reactivity to *Aspergillus fumigatus* in patients with airway inflammation and acute allergic bronchopulmonary aspergillosis [60], establishing a link between protective immunity in the gut and immune pathology in the lung [59,60].

The *C. albicans* transcription factor enhanced filamentous growth 1 (Efg1p) is a major regulator of commensal colonisation in healthy and immune compromised mice. In vivo competition experiments between wild type *C. albicans* and an *efg1*Δ/Δ null mutant strain in healthy (immune competent) mice demonstrate that fungi lacking *EFG1* have an increased propensity to colonise the gastrointestinal tract at early time points (up to 24 h), which is not sustained at later time points [61]. *EFG1* gene expression in wild type *C. albicans* isolated from the caecum or ileum of wild type BALB/c mice was low during the initial stages of gastrointestinal colonisation (within 3 d) but increased over time up to 18 d. In contrast, *EFG1* gene expression in wild type *C. albicans* isolated from T cell-deficient nu/nu mice was consistently low at all time points analysed [61]. Thus, variations in the level of *EFG1* expression within colonising populations of fungi are proposed to reflect an adaptation to the degree of fitness between healthy and compromised hosts, facilitating the transition from commensal to pathogenic behaviour [61].

Indeed, *EFG1* gene dosage has a profound effect on phenotypic plasticity and *C. albicans* commensalism [62]. Many clinical isolates of *C. albicans* are hemizygous for *EFG1* (*EFG1*/*efg1*) and undergo a transition from the white to grey state via a mechanism that involves loss of *EFG1* function, resulting in enhanced fitness in the gastrointestinal tract [62].

*C. albicans* is inordinately responsive to the stresses it frequently encounters and has evolved to use these environmental cues as a means to persist in the host. Passage of *C. albicans* through the murine gastrointestinal tract induces a white-opaque regulator (Wor1p)-dependent switch to the gastrointestinal induced transition (GUT) phenotype that promotes commensalism [63], while *C. albicans* opaque cells exhibit increased colonisation of skin compared to white cells [64].

Very recently, competitive infection experiments between wild type *C. albicans* and a large panel of null mutant strains in a murine model of gastrointestinal colonisation have revealed that the hyphal gene network that promotes virulence causes an inhibition of commensal fitness [55]. Five transcription factor null mutants (*brg1*Δ/Δ, *efg1*Δ/Δ, *rob1*Δ/Δ, *tec1*Δ/Δ and *ume6*Δ/Δ) exhibited enhanced colonisation fitness when compared to an isogenic wild type control strain suggesting that commensal fitness in the gut is inversely related to the expression of genes required for the coordination of morphogenesis [55]. Notably, Ume6p-mediated inhibition of gut colonisation required the activation of the secreted aspartic proteinase Sap6p and, to a lesser extent, Hyr1p [55].

The apparent antagonism between hyphal growth and commensalism is further underpinned by the observation that serial passage of *C. albicans* through the murine gastrointestinal tract promotes adaptive evolution resulting in the loss of hypha-forming ability in the absence of a competitive microbiota [65]. Gut-evolved *C. albicans* strains that lost the ability to form hyphae exhibited reduced virulence in vitro and in vivo. Strikingly, mice “primed” with evolved *C. albicans* received substantial innate protection against systemic infection with *Aspergillus fumigatus*, *Staphylococcus aureus* and *Pseudomonas aeruginosa* [65].

## 5. Invasion of the Mucosal Surface

Mucosal internalisation of *C. albicans* requires a number of host and pathogen-derived factors but remains incompletely understood. There are two main mechanisms of *C. albicans* internalisation. Induced endocytosis describes a process where *C. albicans* remains completely passive during its uptake into host cells, such that metabolically nonviable fungi are still endocytosed. The mechanism is clathrin-dependent and requires actin cytoskeleton remodelling [66]. Induced endocytosis can be instigated by the binding of host E-cadherin or EGFR/Her2 complexes to the fungal invasins Ssa1p or Als3p. Studies show that latex beads coated with the N-terminal region of Als3p are successfully internalised [28], while inhibition of EGFR/Her2 activity significantly reduces the severity of oropharyngeal candidiasis (OPC) [29]. Other proteins with a role in *C. albicans* endocytosis include host GTPases and zonula occludens-1 which are associated with actin remodelling during infection [67], the aryl hydrocarbon receptor (AhR), thought to govern EGFR-induced endocytosis [68], platelet-derived growth factor BB (PDGF BB) and neural precursor cell expressed developmentally downregulated protein 9 (NEDD9), which are important for host-induced uptake of fungus [69]. The major factors that contribute to the process of induced endocytosis of *C. albicans* hyphae are depicted in Figure 1C. 

Active penetration by *C. albicans* results in direct breach of the epithelium, with hyphae extending through individual epithelial cells or between them. Unlike induced endocytosis, this process is dependent on *C. albicans* morphology, where the maintenance of turgor pressure and continued extension of hyphal tips play important roles. Although the presence of *C. albicans* in the gastrointestinal tract is closely associated with commensal carriage, translocation across the gastrointestinal mucosa positively correlates with systemic infection [70,71,72]. Recent research has demonstrated that the predominant route of gastrointestinal translocation by *C. albicans* is transcellular rather than paracellular (occurring through epithelial cells as opposed to between them), a process that is facilitated by the peptide toxin candidalysin [73]. Additionally, fungal secreted aspartic proteases 2 (Sap2p) and Sap5p, which degrade gastrointestinal mucins [74] and E-cadherin [75], respectively, may also facilitate the translocation of *C. albicans* across the epithelium. Though active penetration was considered to be the only mechanism of *C. albicans* internalisation at the gastrointestinal epithelium [76], recent evidence suggests that a host facilitated method may also occur that is dependent on gut M cells [77]. Perhaps unsurprisingly, many of the host proteins that govern fungal invasion possess functions in maintaining the epithelium highlighting the significance of epithelial barrier integrity in protection against *C. albicans* infection [78].

## 6. Epithelial Recognition of *C. albicans*

The first step in developing an innate immune response against *C. albicans* is host recognition. The cells that comprise mucosal barriers express pattern recognition receptors (PRRs) that recognise pathogen-associated molecular patterns (PAMPs) such as fungal cell wall components present on yeast and hyphal cells [79]. There are three main classes of PRR expressed in innate myeloid cells associated with fungal infections: the toll-like receptors (TLRs), the C-type lectin receptors (CLRs) and the NOD-like receptors (NLRs). Of these, TLRs and CLRs are expressed in epithelial cells [80,81,82]. TLRs comprise an extracellular domain rich in leucine repeats required for recognition of microbial structures, and a cytoplasmic Toll/IL-1 receptor (TIR) domain that is responsible for intracellular signalling [80,83,84]. Of all TLRs, only TLR5 and TLR7 are not detectable in human buccal epithelial cells from healthy donors. Epithelial cells interact with *C. albicans* through TLR1-4 and TLR6 [85,86,87,88], with TLR4 being induced during oral candidiasis and TLR2 during vulvovaginal candidiasis [86,89]. 

CLRs recognise polysaccharide structures on microbes and are probably the most important receptors that mediate fungal recognition in macrophages, dendritic cells and neutrophils. Dectin-1 is a β-glucan receptor and constitutes one of the major CLRs in myeloid cells found to be important in systemic infections [90,91,92]. Although dectin-1 can bind to the β-glucan present on yeast phase *C. albicans* [93], infection of TR146 epithelial cells with *C. albicans* for 24 h resulted in a downregulation of dectin-1 gene expression [94]. Indeed, dectin-1 is thought to play only a minor role in the oral epithelial response to *C. albicans* [87]. In contrast, a nonsense mutation in dectin-1 (Y238STOP) is associated with recurrent vulvovaginal candidiasis (RVVC) [95] suggesting a protective role for dectin-1 in the vaginal response to *C. albicans*.

Recently, the ephrin type-A receptor 2 (EphA2) was identified as a nonclassical epithelial PRR that recognises the β-glucans present on *C. albicans* yeast and hyphae [93]. Activation of EphA2 was observed within 15 min in response to yeast phase *C. albicans* and 90 min for hyphae, with subsequent activation of signal transducer and activator of transcription 3 (STAT3), mitogen-activated protein kinase (MAPK) and nuclear factor kappa-light-chain-enhancer of activated B cells (NF-κB) signalling and a proinflammatory and antifungal response at 24 h [93]. These findings suggest a role for EphA2 in the initial interaction between *C. albicans* and epithelial cells, most likely to prime mucosal tissues for subsequent induction of appropriate immune responses to hyphal factors released later during mucosal infection. 

Epithelial cells discriminate between harmless commensalism and invasive hyphae via a biphasic activation of the MAPK immune pathway [96,97]. Recognition of commensal yeast comprises the first phase of the biphasic epithelial response, resulting in a weak activation of the NF-κB, phosphoinositide 3-kinase (PI3K), c-Jun N-terminal kinase (JNK), extracellular signal-regulated kinases (ERK1/2) and p38 MAPK pathways [96,98]. Activation of the NF-κB pathway by fungal cell wall components (e.g., mannan, chitin and β-glucan) is sustained during commensal colonisation, while transient MAPK pathway activity results in the activation of the transcriptional regulator c-Jun in the absence of significant hyphal burdens [96]. Relatively little is known about the epithelial c-Jun response to *C. albicans* yeast, only that it comprises the predominant epithelial response to the yeast morphology of *C. albicans*, and may therefore be considered as a host response to commensal colonisation [96]. Notably, the weak activation of NF-κB, MAPK and PI3K pathways does not result in epithelial damage or the induction of a pro-inflammatory response [96].

In contrast, elevated burdens of invasive hyphae trigger the second phase of the biphasic epithelial response, which is characterised by a strong, sustained activation of the p38, JNK and ERK1/2 MAPK pathways, resulting in c-Fos activation via p38 and the subsequent release of proinflammatory cytokines (GM-CSF, G-CSF, IL-6, IL-1α, IL-1β and IL-36) and the recruitment of innate immune cells, including TCRαβ(+) cells, macrophages and neutrophils [82,96,99,100,101]. The MAPK phosphatase-1 (MKP1) is also activated via MEK1/2-ERK1/2 and acts as a negative regulator of MAPK signalling by dephosphorylating p38 and JNK [98,102]. Together, the ERK/MKP1 and p38/c-Fos signalling pathways alert surrounding tissues to the presence of invasive *C. albicans* hyphae*,* a process which has come to be known as the “danger response” [96]. 

Recently, it was demonstrated that the activation of the epithelial danger response and the damage caused to epithelial cells during *C. albicans* infection is driven by candidalysin; a cytolytic peptide toxin secreted from *C. albicans* hyphae [103,104]. Candidalysin is encoded by the extent of cell elongation 1 (*ECE1*) gene and is derived from sequential proteolytic cleavage of its parent protein Ece1p by kexin-like proteinases [103,105]. *C. albicans* mutants lacking *ECE1* or candidalysin form hyphae and penetrate epithelial cells normally, but do not activate a pro-inflammatory response or cause epithelial damage. During mucosal infection, it is proposed that candidalysin accumulates in an invasion “pocket” created by the invading hyphae. Once the concentration of candidalysin is sufficiently high it causes calcium influx, release of lactate dehydrogenase and membrane destabilisation; all of which are characteristics of several microbial toxins [103,106]. Thus, while epithelial recognition of *C. albicans* β-glucan is mediated through the nonclassical PRR EphA2 [93], it is the recognition of candidalysin activity that drives the host proinflammatory response [103]. The epithelial receptors with known ligands involved in the recognition of *C. albicans* are presented in Table 1.

Approximately 75% of females will experience an episode of vulvovaginal candidiasis (VVC) in their lifetime [107], while approximately 9% will suffer from recurrent VVC [108]. The symptoms of VVC (itching, burning, pain and discharge) are associated with the recruitment of neutrophils into the vaginal lumen [109]. Recent advances have demonstrated that the immune pathology associated with VVC is driven by the secretion of candidalysin from *C. albicans* hyphae [104] and the presence of heparan sulphate, which blocks the interaction between neutrophil Mac1 and *C. albicans* Pra1p required for fungal killing [110,111].

## 7. Epithelial Responses to *C. albicans*

Mucosal surfaces secrete numerous host defence peptides as part of the innate immune response to *C. albicans*. The most prominent of these include the defensins, cathelicidin, lactoferrin, histatin-5 and the alarmins S100A8 and S100A9 [112,113,114]. Defensins are cysteine-rich peptides that can be divided into two families: the α-defensins, which are mainly secreted by neutrophils, and the β-defensins, which are produced from epithelial cells [115]. Recently, however, α-defensin six secreted by human intestinal epithelial cells has been shown to inhibit *C. albicans* invasion and biofilm formation [116]. In humans, four β-defensins are expressed in epithelial cells: hBD-1, hBD-2, hBD-3 and hBD-4 [115,117]. While hBD-1 and hBD-4 are constitutively expressed, hBD-2 and hBD-3 are present at low concentrations and are strongly induced in response to infection or stress [115]. HBD-2 and hBD-3 are found in the human buccal epithelium [118] and are preferentially induced by *C. albicans* hyphae rather than yeast cells [119]. The antifungal activity of both of these defensins requires binding to the fungal invasin Ssa1p [120]. Defensins can also act as chemoattractants for T cells, DCs and neutrophils [120]. Mice deficient in IL-17RA express reduced levels of mBD3 and consequently develop severe OPC [121], underlining the importance of these molecules in the innate response to fungal infection.

LL-37 is the only human cathelicidin identified to date and has a broad spectrum of immune functions including antibacterial action and the ability to induce chemokines. It is a cationic antimicrobial peptide produced by many types of cells including epithelial cells [117]. LL-37 is present in the oral cavity where it inhibits the adherence of *C. albicans* to epithelial cells by interacting with fungal cell wall components such as mannan, chitin and glucan [122]. 

The S100 alarmins are typically found in the cytoplasm but are released into the extracellular environment following tissue damage. A variety of cell types express S100 alarmins including polymorphonuclear neutrophils, monocytes and epithelial cells. The S100 alarmins produced by vaginal epithelial cells are implicated in the recruitment of innate cells following interaction with *C. albicans* [123,124]. Calprotectin, which constitutes a dimer of S100A8 and S100A9, inhibits *C. albicans* cell growth [125,126]. Lactoferrin also possesses antifungal activity by disrupting the *C. albicans* plasma membrane [127] and inducing iron deprivation [128] within the fungus. Histatin-5 is a histidine-rich cationic peptide secreted by human salivary glands. This peptide interacts with the β-glucans present in the *C. albicans* cell wall, binds to the heat shock proteins Ssa1p/2p [129], perturbs the fungal plasma membrane and is translocated into the cytoplasm where it disturbs the balance of cellular ions leading to toxicity [130]. 

Recent research has demonstrated that epithelial cells express the IL-17 receptor (IL-17R) which binds to IL-17 secreted by multiple lymphoid cells including γδ-T, natural killer T (NKT), innate lymphoid cell type 3 (ILC3) and TCRβ+ ‘natural’ Th17 cells (nTh17) [131,132]. From the several IL-17 isoforms, only IL-17A and IL-17F seem to be important in mediating antifungal immunity [114,132]. These isoforms trigger the release of the neutrophil activating chemokine G-CSF and β-defensin-1 and -3 from epithelial cells in oral tissue and the release of histatins from the salivary glands during OPC [112,133]. Mice lacking IL-17R are highly susceptible to OPC [131]. However, human patients deficient in IL-17 secretion or signalling owing to mutations in IL-17F or IL17RA show high susceptibility to mucosal but not invasive candidiasis suggesting that, in humans, Th17 cell responses are mainly necessary for mucosal antifungal responses [134,135,136]. Factors involved in the epithelial response to *C. albicans* are summarised in Table 2.

## 8. Secreted Fungal Factors

In order to persist in the host *C. albicans* must overcome (or avoid) the host immune response and the various challenges posed by the host niche. *C. albicans* secretes 225 proteins which facilitate tissue invasion, immune evasion, nutrient acquisition and organ damage [137]. Many of these secreted proteins are enzymes including lipases, phospholipases and Saps [138], which have different substrate specificities and pH optimums [139]. The best studied of these secreted enzymes are the Saps, which comprise Sap1p to Sap10p [137,140]. Sap1–8p are secreted into the extracellular milieu, while Sap9p and Sap10p remain anchored on the cell membrane [141]. Sap2p and Sap6p stimulate neutrophil chemotaxis during vaginitis [142].

Collectively, the Saps degrade a wide range of host factors including E-cadherin and several involved in the innate and adaptive immune responses (e.g., complement, histatin-5 and antibodies), allowing *C. albicans* to combat host immune defences [140,143]. The amyloidogenic regions within Sap6p contribute to fungal aggregation by binding to zinc and Zap1p-regulated proteins on the hypha surface [144]. Conflicting reports surround the ability of Saps to cause damage to the oral epithelium; while the application of a Sap inhibitor to *C. albicans* reduced levels of epithelial damage [145], studies with Sap deletion mutant strains did not recapitulate this effect [146]. 

Although relatively little is known about secreted lipases and phospholipases in comparison to the Saps, both classes of enzyme are associated with *C. albicans* virulence. Expression of phospholipases is positively correlated with increased epithelial adherence and pathogenicity [147,148]. Notably, secretion of phospholipase B1 (Plb1p) contributes to epithelial penetrance and gastrointestinal translocation [149,150], while a *C. albicans pld1* null mutant forms hyphae in vivo, can adhere to and colonise the murine alimentary tract, but is unable to penetrate the epithelium [151].

*C. albicans* encodes ten lipases designated Lip1p [152] and Lip2-10p [153], which are differentially expressed during mucosal colonisation and infection [154]. Expression of *LIP1-10* is detectable in a reconstituted human oral epithelium infection model after 48 h [155] while analysis of *C. albicans* isolated from the saliva of oral candidiasis patients demonstrates that all lipases except *LIP10* were expressed across the patient cohort [155].

The host mucosa secretes an array of antimicrobial compounds that function to clear invading pathogens. In response to this secreted armoury, *C. albicans* can also produce an arsenal of factors to protect itself from this secreted host response. While not secreted per se, Msb2p is a *C. albicans* cell surface protein that is shed into the extracellular environment where it confers broad-range protection against numerous antimicrobials including LL-37, hBD-1 and histatin-5 [156]. Histatin-5 is also degraded by Sap9p [157] and is actively effluxed from the fungal cell by the polyamine transporter Flu1p [158]. The secreted factors that contribute to *C. albicans* pathogenicity are depicted in Figure 1D.

## 9. Acquisition of Micronutrients

While often discussed in the context of systemic infection, microbial acquisition of host micronutrients is also important at mucosal surfaces. The concentration of zinc, copper and iron in whole saliva is highly variable between individuals and is influenced by numerous factors including diet, gender, age, general health and lifestyle choices such as smoking [159,160,161,162].

Although the gastrointestinal tract receives a steady supply of dietary iron, micronutrients such as zinc are subject to diet-independent changes in availability, for instance, by neutrophil mediated calprotectin-dependent sequestration during gastrointestinal inflammation [163]. Relatively little is known about the concentration of micronutrients at the vaginal mucosa. Throughout the course of host–microbe coevolution, the human host has evolved to withhold specific nutrients from microbes as a defence strategy designed to curtail microbial growth and persistence. To thwart such defences, *C. albicans* has evolved to express and regulate numerous micronutrient acquisition systems in order to persist in the hostile environment of the human body, whether as a commensal or pathogen.

### 9.1. Assimilation of Zinc

Current research is now unveiling the importance of zinc in the complex relationship between *C. albicans* and the human host. Under condition in which nutritional immunity is imposed, *C. albicans* remains capable of acquiring zinc from the host. The pH-regulated antigen-1 (Pra1p) is a secreted zinc-binding “zincophore” that harvests zinc from the host environment before reassociating with the fungal cell and the coexpressed zinc zip-transporter Zrt1p [164]. Sap6p can also bind zinc and is required for zinc uptake and fungal growth in low zinc environments [144].

Zinc homeostasis in *C. albicans* is regulated by the transcriptional activator Zap1p which controls the expression of several genes including the zinc transporters *ZRT1-3* and the vacuolar zinc importer *ZRC1* [165,166]. Conditions of zinc limitation drive the formation of the so-called “Goliath” phenotype, in which *C. albicans* yeast cells become enlarged and exhibit hyper-adherence to polystyrene [167]. *C. albicans* Zrt2p is essential for zinc uptake at acidic pH [168]. Cellular import of zinc is mediated by Zrt1p/Zrt2p while Zrc1p functions during storage of zinc in vesicle-like “zincosomes” [168]. The major factors required for zinc acquisition from the host are depicted in Figure 2A,B.

### 9.2. Iron Uptake

*C. albicans* uses three distinct iron uptake systems; haemoglobin uptake, the reductive iron uptake system and scavenging host siderophores [169,170,171,172]. *C. albicans* can lyse erythrocytes to release haemoglobin from blood [103,173,174,175], which is subsequently bound by the haemoglobin receptor Rbt5p (and its homolog Rbt51p), Pga7p and the secreted haemophore Csa2p [176,177,178,179]. A coordinated haem “acquisition relay” between Rbt5p and Pga7p facilitates the transport of haem from the extracellular environment into vacuoles of the endocytic pathway [179,180], where it can be further processed for use.

*C. albicans* uses a reductive uptake system to acquire iron from ferritin, transferrin [181] and ferric ions. Although a receptor for transferrin has not yet been identified, Als3p exhibits ferritin binding activity in addition to its role as an adhesin and invasin [47]. To utilise the iron found in transferrin, the membrane-associated ferric reductases Cfl1p and Fre10p reduce Fe^3+^ to Fe^2+^ [181,182], which is transported into the cell via a complex comprising an iron permease and a multicopper oxidase [183]. *C. albicans* possesses four iron permeases—the plasma membrane associated Ftr1p and Ftr2p and the vacuolar associated Fth1p and Fth2p [170,183]—and five genes encoding iron ferroxidases, all of which are multicopper oxidases [183,184,185,186]. It is predicted that as many as twenty possible iron transporter complexes may be formed to facilitate iron uptake [183]. The importance of copper in the process of high affinity iron transport is highlighted by gene deletion analysis of the copper-transporting P-type ATPase Ccc2p [187]. A *C. albicans ccc2*Δ/Δ null mutant is unable to use haem as an iron source but may continue to acquire iron from haemin and haemoglobin [187].

*C. albicans* uses the ferrichrome siderophore importer protein Sit1p to scavenge siderophores from bacteria and other fungi. Such behaviour is termed ‘iron parasitism’ and is well documented in numerous microbes [172,188,189]. Sit1p is required for epithelial invasion but is dispensable for disseminated infection in vivo [172]. The major iron uptake systems of *C. albicans* are summarised in Figure 2C–E.

The balance between iron uptake and the avoidance of iron-related toxicity is addressed by an elegant tripartite circuit comprising the transcriptional activator Sef1p which promotes iron uptake together with Sfu1p and Hap34p which repress transcriptional activation of iron-uptake and utilisation genes, respectively [190]. Together, these factors confer resistance to iron depletion in the systemic compartment while enabling the fungus to avoid iron toxicity during commensal colonisation of the gastrointestinal tract [190].

### 9.3. Copper

*C. albicans* uses the transcriptional activator Mac1p to activate expression of the high-affinity copper transporter Ctr1p in copper replete environments [191,192]. Once in the cell, Cu^+^ is rapidly bound to intracellular chaperone proteins before being incorporated into copper requiring proteins. Copper is also required for efficient iron uptake as it is incorporated into the multicopper oxidases of *C. albicans* and a *ctr1* null mutant is unable to grow under conditions of copper or iron limitation [191,192]. *C. albicans* can resist the toxicity associated with the intracellular accumulation of copper ions by activating the copper resistance determinant gene *CRD1* (also known as *CRP1*) encoding a P1-type ATPase copper extrusion pump that removes excess copper ions from the cell [193,194]. A *C. albicans crd1* null mutant is sensitive to exogenous sources of copper, silver and cadmium, suggesting a degree of functional promiscuity towards the efflux of metal ions [193]. 

*C. albicans* Sur7p is a component of the membrane compartment containing Can1 (MCC) plasma membrane subdomain required for appropriate morphogenesis, cell wall synthesis, localisation of actin and responses to cell wall stress [195,196,197,198]. Deletion of *SUR7* results in an increased sensitivity to copper [199]. More recently, the architecture of the fungal plasma membrane has been shown to play an important role in resistance to copper toxicity. The plasma membrane of *C. albicans pil1*Δ/Δ, *lsp1*Δ/Δ, *rvs161*Δ/Δ, *rvs167*Δ/Δ, *arp2*Δ/Δ and *arp3*Δ/Δ mutants are more readily permeabilised by copper and are more susceptible to copper-mediated killing [200]. Copper uptake from the host is depicted in Figure 2F.

## 10. Conclusions

The mucosal surfaces of the human body have a long-established relationship with *C. albicans* that continues to evolve. Throughout this evolving relationship, fungal adaptation and the mucosal immune response have provided a fulcrum about which the balance between commensalism and pathogenicity is determined. Although research often highlights the importance of individual factors to mucosal commensalism and pathogenesis, it must be stressed that both are multifactorial processes that draw heavily upon the combined biological output of numerous molecules. As research continues to explore the interface between fungal adaptation and the host mucosal response, our understanding of the events that determine commensalism and pathogenicity at mucosal surfaces will become ever clearer.

## Figures and Tables

**Figure 1 pathogens-08-00053-f001:**
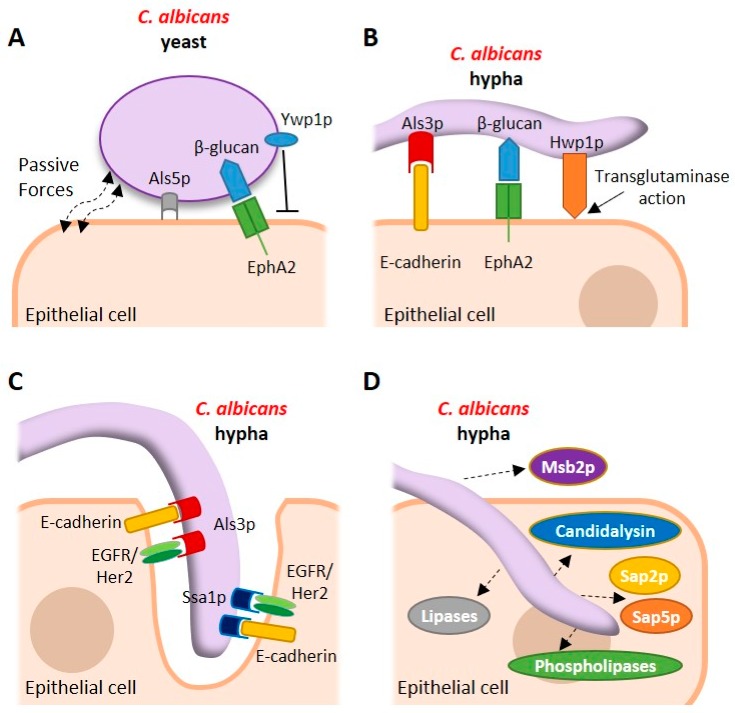
Interactions of *Candida albicans* with host epithelial cells. (**A**) *C. albicans* yeast cells use the passive forces of electrostatic attraction and specific genome-encoded factors such as agglutinin-like sequence 5 (Als5p) to adhere to epithelial cells. Yeast-phase beta glucan is recognised by the nonclassical pattern recognition receptor EphA2 during this initial interaction. Ywp1p is expressed during yeast-phase growth and has antiadhesive properties. A host receptor for Ywp1p has not yet been identified. (**B**) Once attached to the mucosal surface the transition to the hyphal morphology results in the expression of additional adhesins including Als3p and Hwp1p, which further consolidate epithelial adhesion by interacting with E-cadherin and acting as a substrate for host transglutaminase enzymes, respectively. Hyphal beta glucan is also recognised by EphA2 during this strengthened adherence. (**C**) Epithelial internalisation of *C. albicans* hyphae is mediated by the Als3p and Ssa1p invasins which interact with E-cadherin and a heterodimeric receptor complex comprising the epidermal growth factor receptor (EGFR) and Her2 (EGFR/Her2). *C. albicans* remains passive during this process of pathogen-induced endocytosis but may also breach mucosal barriers directly by active penetration. (**D**) While in contact with the mucosal surface, *C. albicans* secretes an arsenal of virulence factors including the peptide toxin candidalysin, secreted aspartic proteinases (Saps), lipases and phospholipases that facilitate pathogenicity. Msb2p is released into the extracellular environment to counteract the activity of numerous host antimicrobial peptides.

**Figure 2 pathogens-08-00053-f002:**
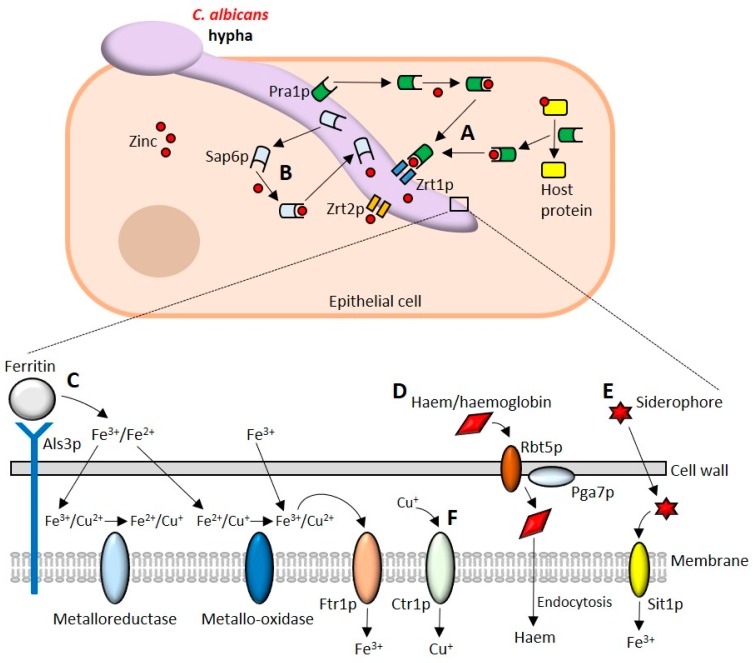
Acquisition of micronutrients by *Candida albicans*. (**A**) *C. albicans* uses the secreted zincophore Pra1p to bind free zinc and scavenge zinc from zinc-containing host proteins. Once zinc is acquired it is imported into the cell by the zinc transporter Zrt1p. Zrt2p is also capable of importing zinc and is essential for zinc uptake at acidic pH. (**B**) The secreted aspartic proteinase Sap6p is also a zincophore capable of binding and importing zinc. (**C**) Als3p can bind to host ferritin to release ferric and ferrous ions that are reduced and oxidised by ferric reductases and ferric oxidases, respectively. *C. albicans* can also acquire free iron which is imported into the cell by Ftr1p. (**D**) The haemolytic activity of *C. albicans* liberates haemoglobin from blood. Haemoglobin/haem are bound by the haemoglobin receptor Rbt5p and by Pga7p and endocytosed into the cell. (**E**) *C. albicans* can also acquire iron by scavenging siderophores. The ferrichrome transporter Sit1p is used to import ferric ions. (**F**) Import of copper is mediated by the copper transporter Ctr1p.

**Table 1 pathogens-08-00053-t001:** Epithelial receptor–ligand pairings involved in recognition of *C. albicans*.

Receptor	Ligand	Reference
Aryl hydrocarbon receptor	IFN-γ and L-kynurenine	[68]
Dectin-1	β-glucan	[90]
EGFR/Her2	Als3p/Ssa1p	[28,29]
EphA2	β-glucan	[93]

**Table 2 pathogens-08-00053-t002:** Factors involved in the host epithelial response to *C. albicans*.

Factor	Target	Cell/Tissue	Reference
α-defensin 6	Invasion/biofilm formation	Intestinal ECs	[116]
Murine β-defensin 1	Reduces mucosal infection	Oral cavity	[133]
β-defensin 2	Ssa1p	Buccal epithelium	[118,120]
β-defensin 3	Ssa1p	Buccal epithelium	[118,120]
Cathelicidin	Fungal adherence	Oral cavity	[122]
S100 alarmins	Immune cell recruitment	Vaginal ECs	[123,124]
Calprotectin	Fungal cell growth	In vitro	[125]
Lactoferrin	Plasma membrane	In vitro	[127]
Histatin-5	Ssa1/2p	Salivary gland	[129,130]

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
