# Peer review of "Candida albicans Interactions with Mucosal Surfaces during Health and Disease"

_pathogens, 2019, doi:10.3390/pathogens8020053_

Round 1
Reviewer 1 Report
This useful review nicely summarizes a lot of data on Candida-mucosal interactions. It is clearly written and generally accurate. However, it would be easier to think about all the data if there were cartoon figure(s) or table(s) summarizing the Epithelial Recognition and Epithelial Responses sections (6 and 7).
Minor points
l. 84: Meaning not clear: how can Als2-7 be highly expressed, whereas Als4 and Als5 are not? These numbers are presumably included in “2-7”
l. 132: That Als1p “plays no role” is an overstatement. It plays a more minor role than Als3
l. 189: Data of Gaur, Klotz, and co-workers show signficant binding to peptides with a free N-terminal and bead-bound C-terminals. These data should be acknowledged.
l. 193: Interaction of soluble AFR regions has also been shown by amyloid fiber formation in Als5p (ref 52).
l. 215: It would be clarifying to cite specific data here on whether increased or decreased EFG1 expression facilitates commensal vs. pathogenic behavior
l. 230: why is the reference number underlined?
l. 273: Is CLEC7A the gene for dectin-1?
l. 349: correct to “…, perturbs the fungal…”
Author Response
Reviewer 1
This useful review nicely summarizes a lot of data on Candida-mucosal interactions. It is clearly written and generally accurate. Response: We thank the reviewer for their kind words.
However, it would be easier to think about all the data if there were cartoon figure(s) or table(s) summarizing the Epithelial Recognition and Epithelial Responses sections (6 and 7). Response: We have now included additional tables summarising epithelial recognition of C. albicans and the major factors produced by epithelial cells in response to C. albicans.
Minor points
l. 84: Meaning not clear: how can Als2-7 be highly expressed, whereas Als4 and Als5 are not? These numbers are presumably included in “2-7” Response: We agree with the reviewer and have changed the text accordingly.
l. 132: That Als1p “plays no role” is an overstatement. It plays a more minor role than Als3 Response: The authors agree with the reviewers comment and have changed the text accordingly.
l. 189: Data of Gaur, Klotz, and co-workers show signficant binding to peptides with a free N-terminal and bead-bound C-terminals. These data should be acknowledged. Response: We thank the reviewer for pointing this out and have now included the relevant citations.
l. 193: Interaction of soluble AFR regions has also been shown by amyloid fiber formation in Als5p (ref 52). Response: We have now included a sentence describing the formation of amyloid fibers by the AFR of Als5p.
l. 215: It would be clarifying to cite specific data here on whether increased or decreased EFG1 expression facilitates commensal vs. pathogenic behavior Response: We have now summarised the findings of in vivo competition experiments between wild type and efg1Δ/Δ null mutant strains and have included further descriptive detail clarifying the role of EFG1 expression during the gastrointestinal colonisation of healthy and immune compromised murine models as described by Pierce et al.
l. 230: why is the reference number underlined? Response: We thank the reviewer for pointing this out. The text has now been changed.
l. 273: Is CLEC7A the gene for dectin-1? Response: CLEC7A is indeed the gene that encodes dectin-1. We have provided additional clarification in the text.
l. 349: correct to “…, perturbs the fungal…” Response: We thank the reviewer for pointing this out. The text has now been corrected.

Reviewer 2 Report
The manuscript by Nikuo, et al is a timely review of the current literature on Candida albicansinteractions with oral epithelial cells. The title is “Candida albicans interactions with mucosal surfaces during health and disease”, however the authors do not cover the current literature on Candida interactions with vaginal epithelium, including exciting work regarding the role of immune-driven pathology in chronic vaginitis, the role of heparan sulfate in mediating this, or the role of candidalysin in this type of pathology. They also focus mostly on epithelial cells and not so much on adaptive immune responses. Thus, I think the title should be changed to focus on oral epithelial cells to reflect this.
A few other points that the authors should address are:
1. Lines 190-200: The section on amyloids states that the amyloids are disrupted by ligand binding, but this seems to be opposite of the reports in the literature which are well-summarized in the review Lipke et al Microbiol Mol Biol Rev. 2017 Nov 29;82(1). pii: e00035-17.
2. Lines 211-215: The section on Efg1 would benefit from more detail. For example, describe how expression levels of Efg1 affect colonization rather than just stating that they correlate.
3. What are the acronyms used as short hand for these factors in lines 233 and 234? platelet derived growth factor BB and neural precursor cell expressed developmentally down-regulated protein 9
4. Line 273-274: If the authors focus on oral epithelium this is probably fine, but if they cover vaginal epithelium then they should describe the data from Ferwerda et al N Engl J Med 2009; 361:1760-1767, which implicate dectin-1 in playing a role.
5. Figure 2D: I think the authors should include Pga7 in this cartoon since they describe Rbt5 and its role in the haem relay.
6. Lines 485-495: For the section on copper, the very recent article Douglas & Konopka PLoS Genet. 2019 Jan 11;15(1):e1007911 should be included.
Author Response
Reviewer 2
The manuscript by Nikuo, et al is a timely review of the current literature on Candida albicansinteractions with oral epithelial cells. The title is “Candida albicans interactions with mucosal surfaces during health and disease”, however the authors do not cover the current literature on Candida interactions with vaginal epithelium, including exciting work regarding the role of immune-driven pathology in chronic vaginitis, the role of heparan sulfate in mediating this, or the role of candidalysin in this type of pathology. They also focus mostly on epithelial cells and not so much on adaptive immune responses. Thus, I think the title should be changed to focus on oral epithelial cells to reflect this. Response: We thank the reviewer for bringing this omission to our attention. We have now included a short section in the manuscript that describes the involvement of candidalysin in driving immune pathology, and the recent advances concerning the role of heparan sulphate in the induction of neutrophil anergy. In addition to these inclusions, the revised manuscript also contains additional material that explore the relationship between C. albicans and the gastrointestinal mucosa. We agree that the manuscript does not focus on the adaptive immune response to C. albicans, but on the interaction of C. albicans with mucosal surfaces as described in the manuscript title. An in-depth discussion of the adaptive immune response to C. albicans is beyond the scope of this paper.
A few other points that the authors should address are:
1. Lines 190-200: The section on amyloids states that the amyloids are disrupted by ligand binding, but this seems to be opposite of the reports in the literature which are well-summarized in the review Lipke et al Microbiol Mol Biol Rev. 2017 Nov 29;82(1). pii: e00035-17. Response: The text in our manuscript refers to peptide ligands that interact directly with the peptide binding cavity (PBC). In contrast, the ligands referred to in the Lipke review have not been yet structurally characterised and their binding pocket has yet to be identified. To provide clarity to this issue, we have changed the text in the manuscript to ' ... in the absence of ligands. However, in the presence of ligands that bind to the PBC these interactions are disrupted. '
2. Lines 211-215: The section on Efg1 would benefit from more detail. For example, describe how expression levels of Efg1 affect colonization rather than just stating that they correlate. Response: We have now summarised the findings of in vivo competition experiments between wild type and efg1Δ/Δ null mutant strains and have included further detail clarifying the role of EFG1 expression during the gastrointestinal colonisation of healthy and immune compromised murine models as described by Pierce et al.
3. What are the acronyms used as short hand for these factors in lines 233 and 234? platelet derived growth factor BB and neural precursor cell expressed developmentally down-regulated protein 9 Response: The acronyms have now been included in the text as requested.
4. Line 273-274: If the authors focus on oral epithelium this is probably fine, but if they cover vaginal epithelium then they should describe the data from Ferwerda et al N Engl J Med 2009; 361:1760-1767, which implicate dectin-1 in playing a role. Response: We have now included this citation in the manuscript.
5. Figure 2D: I think the authors should include Pga7 in this cartoon since they describe Rbt5 and its role in the haem relay. Response: We have now included Pga7p in Figure 2D.
6. Lines 485-495: For the section on copper, the very recent article Douglas & Konopka PLoS Genet. 2019 Jan 11;15(1):e1007911 should be included. Response: We thank the reviewer for bringing this recent (and important) paper to our attention. We have now included a summary of the major findings of Douglas and Konopka together with a brief additional statement regarding the importance of Sur7p in copper resistance.

Reviewer 3 Report
The interaction of the opportunistic fungal pathogen Candida albicans with the host mucosal surfaces is essential for the commensal lifestyle of the fungus, and represents the potential gateway for tissue invasion leading to pathogenesis. This comprehensive, clear and well-written review on this topic constitutes a valuable contribution to the field. I have just a couple of suggestions for the authors' consideration, and a few specific comments.
1) A series of papers came out very recently - some probably after the submission of the manuscript - that underscore the antagonistic role between the hyphal gene expression program and persistence as a commensal in the mouse gut (Tso 2018, DOI: 10.1126/science.aat0537and Liang 2019, https://doi.org/10.1016/j.chom.2019.01.005), and specifically between SAP6 expression and GI persistence (Witchley 2019, https://doi.org/10.1016/j.chom.2019.02.008). Another recent report (Shao 2019,DOI:https://doi.org/10.1016/j.chom.2019.02.004) as well as Tso 2018 further find that C. albicans colonization modulates the mouse immune response towards Candida as well as other pathobionts. These reports seem to me to be very pertinent to the topic of the review.
2) The section on nutrient acquisition is a welcome addition to this microorganism - mucosal surfaces interactions review. However, I believe that some essential background is lacking: since micronutrient availability is usually discussed in the context of systemic infection, it would be helpful to describe what is known about micronutrient availability in the mucosal environments. Iron might be limiting due to lactoferrin, but what about the other metals? In the gut, the diet is presumably the principal source of metals (added to some hemoglobin from low level bleeding). What about other mucosal surfaces? Are metals such as copper in excess or in deficit, i.e., is Ctr1 or Crp1 required, and where? etc.
Specific comments:
line 412-414: the statement is about C. albicans zinc regulation by Zap1, but reference 148 is about S. cerevisiae. The correct references for CaZAP1/CSR1 are Nobile, PLOS Biology 2009 and Kim, Microbiol Biotechnol 18: 242–247 2008.
line 457: the correct reference for the secreted hemophore function of Csa2 is Nasser, Nat. Microbiol. 2016, rather than reference 159.
line 464: the correct references for functional analysis in Candida of ferric reductase activities that are involved in free iron and transferrin iron (not ferritin iron as far as I am aware) uptake are ref 163, Knight 2005 (Fre10), and ref 165 (Cfl1) rather than ref. 164, which contains no functional analyses.
line 467: ref. 167 is on S. cerevisiae, not C. albicans vacuolar iron transport. The correct references for the preceding statements are 152 and 166.
line 470: It could be argued that the ccc2 mutant phenotype highlights the importance of copper in iron transport (for biogenesis of all five copper ferroxidases, presumably), rather than the importance of iron per se.
line 490: the correct references for the preceding statements are 175, 176 but not 169.
Typos:
line 377: "effect"?
Author Response
Reviewer 3
The interaction of the opportunistic fungal pathogen Candida albicans with the host mucosal surfaces is essential for the commensal lifestyle of the fungus, and represents the potential gateway for tissue invasion leading to pathogenesis. This comprehensive, clear and well-written review on this topic constitutes a valuable contribution to the field. Response: We thank the reviewer for their positive feedback.
I have just a couple of suggestions for the authors' consideration, and a few specific comments.
1) A series of papers came out very recently - some probably after the submission of the manuscript - that underscore the antagonistic role between the hyphal gene expression program and persistence as a commensal in the mouse gut (Tso 2018, DOI: 10.1126/science.aat0537and Liang 2019, https://doi.org/10.1016/j.chom.2019.01.005), and specifically between SAP6 expression and GI persistence (Witchley 2019, https://doi.org/10.1016/j.chom.2019.02.008). Another recent report (Shao 2019,DOI:https://doi.org/10.1016/j.chom.2019.02.004) as well as Tso 2018 further find that C. albicans colonization modulates the mouse immune response towards Candida as well as other pathobionts. These reports seem to me to be very pertinent to the topic of the review. Response: We thank the reviewer for bringing these important papers to our attention and have included the major findings of each study in the revised manuscript.
2) The section on nutrient acquisition is a welcome addition to this microorganism - mucosal surfaces interactions review. However, I believe that some essential background is lacking: since micronutrient availability is usually discussed in the context of systemic infection, it would be helpful to describe what is known about micronutrient availability in the mucosal environments. Iron might be limiting due to lactoferrin, but what about the other metals? In the gut, the diet is presumably the principal source of metals (added to some hemoglobin from low level bleeding). What about other mucosal surfaces? Are metals such as copper in excess or in deficit, i.e., is Ctr1 or Crp1 required, and where? etc. Response: The reviewer raises an interesting question. The amount of micronutrients available at mucosal surfaces is highly variable and influenced by several factors in addition to diet (for example gender, age, exercise together with lifestyle choices such as smoking) and mechanisms such as calprotectin-mediated sequestration of zinc during mucosal inflammation. Only a handful of studies have attempted to quantify the concentration of micronutrients at mucosal surfaces, and there is clear discordance between published data, most likely because of the variables described. However, the authors agree that some discussion of micronutrient levels is warranted and have included a short statement regarding the limited number of studies that have attempted to quantify micronutrient levels and the variations observed between datasets.
Specific comments:
line 412-414: the statement is about C. albicans zinc regulation by Zap1, but reference 148 is about S. cerevisiae. The correct references for CaZAP1/CSR1 are Nobile, PLOS Biology 2009 and Kim, Microbiol Biotechnol 18: 242–247 2008. Response: We thank the reviewer for bringing this to our attention. The correct citations have now been included.
line 457: the correct reference for the secreted hemophore function of Csa2 is Nasser, Nat. Microbiol. 2016, rather than reference 159. Response: The correct citation has now been inserted.
line 464: the correct references for functional analysis in Candida of ferric reductase activities that are involved in free iron and transferrin iron (not ferritin iron as far as I am aware) uptake are ref 163, Knight 2005 (Fre10), and ref 165 (Cfl1) rather than ref. 164, which contains no functional analyses. Response: We thank the reviewer for bringing this to our attention. We have now included the correct citations.
line 467: ref. 167 is on S. cerevisiae, not C. albicans vacuolar iron transport. The correct references for the preceding statements are 152 and 166. Response: We thank the reviewer for bringing this oversight to our attention. The citations have now been corrected.
line 470: It could be argued that the ccc2 mutant phenotype highlights the importance of copper in iron transport (for biogenesis of all five copper ferroxidases, presumably), rather than the importance of iron per se. Response: We thank the reviewer for pointing this out. We have now included additional description in the text describing the role of Ccc2p in iron transport.
line 490: the correct references for the preceding statements are 175, 176 but not 169. Response: We thank the reviewer for bringing this oversight to our attention. The citations have now been corrected.
Typos:
line 377: "effect"? Response: This has now been changed.
